# Neuron Compatibility and Antioxidant Activity of Barium Titanate and Lithium Niobate Nanoparticles

**DOI:** 10.3390/ijms23031761

**Published:** 2022-02-03

**Authors:** Mariarita Candito, Edi Simoni, Erica Gentilin, Alessandro Martini, Gino Marioni, Serena Danti, Laura Astolfi

**Affiliations:** 1Bioacoustics Research Laboratory, Department of Neurosciences, University of Padova, via G. Orus, 2b, 35129 Padova, Italy; mariarita.candito@phd.unipd.it (M.C.); edi.simoni@unipd.it (E.S.); erica.gentilin@unipd.it (E.G.); alessandromartini@unipd.it (A.M.); 2National Interuniversity Consortium of Materials Science and Technology (INSTM), via G. Giusti 9, 50121 Firenze, Italy; serena.danti@unipi.it; 3I–APPROVE, International Auditory Processing Project in Venice, Department of Neurosciences, University of Padova, Santi Giovanni e Paolo Hospital, ULSS3 Serenissima, 30122 Venezia, Italy; 4Otolaryngology Unit, Department of Neuroscience DNS, University Hospital of Padova, via Giustiniani, 2, 35129 Padova, Italy; 5Department of Civil and Industrial Engineering, University of Pisa, Largo Lucio Lazzarino, 56126 Pisa, Italy; 6Department of Civil and Environmental Engineering, Massachusetts Institute of Technology (MIT), Massachusetts Ave. 77, Cambridge, MA 02139, USA

**Keywords:** apoptosis, neuronal cell line, oxidative stress, PC12, piezoelectric material, ROS

## Abstract

The biocompatibility and the antioxidant activity of barium titanate (BaTiO_3_) and lithium niobate (LiNbO_3_) were investigated on a neuronal cell line, the PC12, to explore the possibility of using piezoelectric nanoparticles in the treatment of inner ear diseases, avoiding damage to neurons, the most delicate and sensitive human cells. The cytocompatibility of the compounds was verified by analysing cell viability, cell morphology, apoptotic markers, oxidative stress and neurite outgrowth. The results showed that BaTiO_3_ and LiNbO_3_ nanoparticles do not affect the viability, morphological features, cytochrome c distribution and production of reactive oxygen species (ROS) by PC12 cells, and stimulate neurite branching. These data suggest the biocompatibility of BaTiO_3_ and LiNbO_3_ nanoparticles, and that they could be suitable candidates to improve the efficiency of new implantable hearing devices without damaging the neuronal cells.

## 1. Introduction

Recent investigations in nanotechnology, focused on small-scale compounds using multidisciplinary approaches (e.g., medicine, biology and bioengineering), have allowed for significant technological advancements, paving the way to achieve challenging objectives [1,2]. Among the compounds currently under consideration, micro- and nanoparticles play an important role due to their size, which is similar to cellular and subcellular structures, making them capable of interacting with biological processes that are unattainable by macro- and micro-devices [3,4,5,6]. Nanoparticles can be modulated to cross the metabolic barriers; thus, they are very important in the treatment of many diseases that are difficult to cure because of the organ they affect, such as the human ear [7,8].

According to World Health Organization, approximately 466 million worldwide suffer from disabling hearing loss [9]. Sensory Neural Hearing Loss (SNHL), the main type of hearing impairment, is a heterogeneous disorder affecting the cochlea or the auditory nerve, with irreversible damage to the inner ear’s sensory cells and neurons. It may be caused by aging, genetic factors, trauma, drug ototoxicity or noise exposure [10,11,12,13,14].

Recovery from SNHL is related to the implantation of a Cochlear Implant (CI) that can stimulate the cochlear nerve, bypassing damaged sensory cells and allowing the brain to receive auditory stimuli [15,16]. The CI helps deaf people to rediscover the joy of sound, but insertion of this device is often accompanied by medical side effects (e.g., infections, electrode migration, pain, inner ear trauma, dizziness) and device failures (e.g., damage to electronic components), which lead to explantation [17,18,19,20].

In recent years, researchers have worked on the development of a new “self-powered” device for cochlear stimulation based on piezoelectric nanomaterials, exploiting the characteristics of these compounds to obtain a CI that can optimize sound reception, keeping it as natural as possible and reducing side effects related to the conventional CI. The device is made of electrospun fibres based on a piezoelectric polymer, polyvinylidene fluoride (PVDF), which is already known to be biocompatible and very flexible, to which lithium niobate (LiNbO_3_) has been added, a nanoceramic with high piezoelectric properties, currently used in bioengineering [21,22].

The present study aims to analyse the cytocompatibility of two piezoelectric nanoparticles, barium titanate (BaTiO_3_) and LiNbO_3_, on a neuronal cell line derived from a rat pheochromocytoma of the adrenal medulla (PC12).

Innovative use of piezoelectric materials in the production of new CI electrodes allows for economical, biomimetic, water and magneto-compatible devices to be obtained, which could drastically improve the quality of life for people affected by SNHL [21,22]. The state of the spiral ganglion neurons, namely, the target cells of the cochlear implant stimulation is essential to the proper functioning of a cochlear implant [15,16,17,18,19,20]. For this reason, studying the safety of the materials used to produce CI electrodes on nerve fibres is pivotal.

Barium titanate is a piezoelectric ceramic characterized by a high dielectric constant and good biocompatibility. It is currently used in nanomedicine for drug delivery or tissue engineering [23,24], and for the construction of new, smart piezoelectric membranes, which can induce the regeneration of osteoarticular tissues [25,26]. Lithium niobate is another piezoelectric ceramic material characterized by a high spontaneous polarization and large pyroelectric, electro-optical and photoelastic coefficients [27,28].

Due to its properties, LiNbO_3_ is widely used in many applications, such as acoustic wave transducers, optical phase modulators and memory elements. Moreover, LiNbO_3_ nanoparticles can stimulate cell adhesion and proliferation and are currently under investigation to improve the regeneration and healing of tissues [21,22,27,28,29].

Previous data obtained by testing LiNbO_3_ cytocompatibility on inner ear and neuronal cells showed encouraging results [21]. The present study analyses the neurotoxic effects and the antioxidant activity of piezoelectric nanoparticles in differentiated PC12 cells. The PC12 are chromaffin cells derived from rat pheochromocytoma, which can differentiate into neuron-like cells when stimulated with Nerve Growth Factor (NGF). Once differentiated, PC12 cells stop proliferating, extend their neurites and axons, and exhibit the morphological features of sympathetic neurons. For this reason, they are widely used in neurotoxicity studies regarding neuroprotection and neuroinflammation [30,31], and as a neurocompatibility model for inner ear device development [32,33]. The results obtained in this study could provide new information on the possible use of piezoelectric nanoparticles in the treatment of inner ear diseases, avoiding damage to the most fragile human cells, the neurons.

## 2. Results

### 2.1. Cell Viability Assay

Barium titanate (BaTiO_3_) and lithium niobate (LiNbO_3_) had a submicrometric size, as detected via scanning electron microscopy (SEM) analysis (Appendix A). To investigate the cytocompatibility of BaTiO_3_ and LiNbO_3_ nanoparticles on the PC12 cell line, the nanoparticles were treated according to an established procedure to generate a uniform dispersion [21]. Cell viability was assessed by MTS assay. The results (Figure 1a) showed that BaTiO_3,_ at concentrations 5.8 nM after 24 h, 500 nM after 24 and 72 h, and 100 nM at all times tested, significantly increased PC12 cells’ viability. Similarly, the treatment with LiNbO_3_ significantly increased cell viability after 24 h at all concentrations tested (Figure 1b). The same effect was observed after 48 and 72 h at the highest concentrations tested. On the contrary, the treatment with 13 µM cisplatin at all times tested, and 25 µM after 24 h, significantly decreased PC12 cell viability. These results show that treatment with piezoelectric nanoparticles not only did not alter the cell viability of differentiated PC12 cells, but also improved their proliferation rate.

### 2.2. Cell Morphology

To evaluate the effects of nanoparticles on cell morphology, treated cells were analysed using the phalloidin TRITC/DAPI staining (Figure 2a–x). The treatment with BaTiO_3_ did not alter the morphological features of differentiated PC12 cells at any of the times or doses tested (Figure 2d–l); actin filaments in the cytoskeleton were normally distributed and the nuclei were rounded without signs of pyknosis. Similarly, the treatment with LiNbO_3_ did not affect the morphology of PC12 cells at any of the times and doses tested (Figure 2m–u). Conversely, the positive control induced neurotoxic effects on PC12 cells (Figure 2v–x), including a decreased mean size of cytoskeleton elements, cell shrinkage, multinucleated cells and “destructive fragmentation of nuclei”.

### 2.3. Cytochrome c Expression by PC12 Cells

To evaluate the effects of piezoelectric nanoparticles on cytochrome c expression, treated cells were analysed by immunocytochemistry (Figure 3a–x). The results showed that treatment with BaTiO_3_ at all times and doses tested did not alter the cytochrome c distribution, in comparison with untreated PC12 cells. Similarly, treatment with LiNbO_3_ at all times and doses tested showed a homogenous distribution of cytochrome c (Figure 3d–u). The positive control caused neurotoxic effects and diffused cytochrome c staining [34] within the cytoplasm (Figure 3v–x).

### 2.4. ROS Production by PC12 Cells

The ROS production in PC12 cells treated with nanoparticles was evaluated by H_2_DCFDA assay. The results showed that treatment with BaTiO_3_ did not significantly affect the levels of ROS produced by PC12 in comparison to untreated cells (Figure 4a). Similarly, the treatment with LiNbO_3_ nanoparticles did not increase ROS production in PC12 cells and significantly reduced it at 100 nM and 24 h of treatment (Figure 4b). Thus, barium titanate and lithium niobate nanoparticles did not significantly induce oxidative stress. The positive control significantly increased ROS levels by 40% compared to untreated cells.

### 2.5. Neurite Outgrowth

Finally, to evaluate the neuromodulatory effects of piezoelectric nanoparticles, the neurite outgrowth was analysed on treated PC12 cells based on the parameters described in the Section 4.

Differentiation grade (Figure 5a). The treatment with BaTiO_3_ did not affect the cell differentiation grade at any of the times and doses tested. The treatment with LiNbO_3_ did not reduce the number of differentiated cells in comparison with untreated cells at any of the times and doses tested. Conversely, treatment with the positive control (13 µM cisplatin) induced a significant reduction in this parameter, starting 24 h after treatment.

Number of neurites per cell (Figure 5b). The treatment with BaTiO_3_ did not alter the number of neurites produced by treated cells at any of the times and doses tested, and the same effect was observed on cells treated with LiNbO_3_. Conversely, the treatment with the positive control significantly reduced this parameter, starting 48 h after treatment.

Average length of neurites (Figure 5c). The treatment with BaTiO_3_ did not affect the average length of neurites, but the length increased at 5.8 nM after 24 h and at 100 nM after 48 h. Similarly, the treatment with LiNbO_3_ did not reduce this parameter but significantly increased it at 5.8 nM after 24 h. The treatment with positive control significantly reduced neurite length after 48 h of treatment.

Number of branch points (Figure 5d). The treatment with BaTiO_3_ significantly increased the number of branch points produced by PC12 cells at 500 nM, starting 48 h after treatment. Initially, the treatment with BaTiO_3_ at 5.8 nM significantly reduced the number of branches after 24 h, but 72 h after treatment, this parameter significantly increased. Treatment with LiNbO_3_ significantly increased the number of branch points at the higher concentrations tested (100 nM and 500 nM) 48 h after treatment, and at 5.8 and 100 nM, 72 h after treatment. The treatment with positive control significantly reduced the number of branch points produced by PC12 cells at all times tested.

## 3. Discussion

The cellular endogenous electric field influences tissue regeneration and regulates cellular processes such as migration, chemiotaxis, proliferation, intracellular communication and neuronal activity. Many studies involve the use of electricity to accelerate healing processes, bone regeneration and neuronal stimulation [35]. The materials that attract the most interest are the piezoelectric ones: under an external and wireless mechanical stimulus, for example, ultrasonic stimulation or vibration, they are able to produce an electric field and transmit it to the cells [36]. These materials are already used as scaffolds in many research fields, due to their biocompatibility and ability to stimulate tissue regeneration [21,22,25,37,38]. It has been shown that, by producing an electric field, these materials can stimulate neuronal extension and proliferation, bone formation and repair, and cell migration [39]. The aim of this work was to analyse the cytocompatibility of two piezoelectric ceramic nanoparticles, barium titanate (BaTiO_3_) and lithium niobate (LiNbO_3_), on a neuronal cell line to investigate the possible use of these nanoparticles for the construction of new self-powered cochlear implants (CI). This innovative device could be a turning point in improving the quality of life of people with deafness, but first the biocompatibility of these nanomaterials should be investigated in detail to avoid possible side effects for neurons.

Many studies describe BaTiO_3_ as useful not only in electronic devices [40], but also in biomedicine for drug delivery systems [41] and wireless neuronal stimulation [42]. In previous studies, we have established methodologies for obtaining suitable dispersions of these ceramic nanoparticles, which make them suitable for cellular uptake [21,42,43]. Barium titanate is widely used as a piezoelectric ceramic, and its biocompatibility has been investigated on many different cell lines [24]. The results obtained in this study show that BaTiO_3_ nanoparticles significantly increase the viability of PC12 cells. These data agree with previous studies that analysed the bioactivity of glycol-chitosan coated BaTiO_3_ nanoparticles on the human neuroblastoma SH-SYY cell line and mesenchymal stem cells, respectively, showing a high viability and cell proliferation [42,44]. Moreover, treatment with BaTiO_3_ did not affect cell morphology, cytochrome c distribution and the amount of ROS produced by differentiated PC12 cells, confirming that these nanoparticles did not enhance cell oxidative stress, as previously found on mesenchymal stem cells [44].

Concerning neurite outgrowth, the treatment with BaTiO_3_ did not affect the differentiation grade and the neurite number and length in treated cells at any of the times and doses tested. On the contrary, the neurite length was significantly increased at 5.8 nM, 24 h after treatment, and at 100 nM, 48 h after treatment. These data agree with those obtained in a previous study [45] that analysed the effect of other nanoparticles, the boron nitride nanotubes (BNNT) on differentiated PC12 cells, observing an increase in neurite length and number.

In this study, the neurite branching of PC12 cells, a key aspect for neuronal structure [41], was significantly enhanced by BaTiO_3_ nanoparticles at the highest concentrations tested (500 nM), starting 48 h after treatment. The smallest concentration of BaTiO_3_ (5.8 nM) caused a significant reduction in the branch points produced by cells 24 h after treatment; however, after 72 h, this parameter was significantly enhanced. These data suggest that BaTiO_3_ nanoparticles stimulate an increase in the complexity of the neurite network produced by PC12 cells. They are also consistent with the results obtained in a previous study, observing an increase in the high-amplitude Ca^2+^ transients on SH-SY5Y-derived neurons treated with BaTiO_3_ nanoparticles and stimulated with ultrasounds [46]. The calcium flux stimulates the neuronal differentiation and the neurite outgrowth; thus, this could be the reason that piezoelectric nanoparticles have neuromodulatory effects. These properties make them suitable candidates for wireless neuronal stimulation [46]. Moreover, another study showed that PC12 cells treated with ZnO and TiO_2_ nanoparticles, which are cytotoxic for this cell line, produce a lower number of neurites and of branches [47]. These data, together with the results obtained in this study from the treatment at the highest concentration of piezoelectric nanoparticles, suggest that the BaTiO_3_ is biocompatible towards the PC12 cell line.

The LiNbO_3_ is a piezoelectric ceramic, currently employed for its properties in many research fields, for example, in the opto-electronic industry [48], as transducers to generate surface acoustic waves [49], in tissue engineering [28,29] and in bio-imaging [50,51]. However, to date, few studies have examined the cytocompatibility of LiNbO_3_. The present study analysed the effects of LiNbO_3_ nanoparticles on the selected neuronal cell line, PC12. The results showed that LiNbO_3_ did not affect the morphological features of PC12 cells, in agreement with a previous study that analysed the effects of osteoblast cells (MC3T3) seeded onto LiNbO_3_ crystal surface [29]. The charged surface of LiNbO_3_ enhanced the adhesion and proliferation of cells without modifying the cell’s morphological features. In the same way, Marchesano and collaborators [28], and Li and collaborators [52] described the effects of mouse embryonic fibroblasts (NIH-3t3) and rat bone marrow mesenchymal stem cells (rBMMSCs) seeded onto the surface of LiNbO_3_ crystals, showing increased adhesion, proliferation and cell migration. In the present study, the LiNbO_3_ nanoparticles did not affect cytochrome c distribution or the amount of ROS produced by PC12 cells, thus suggesting a high in vitro compatibility of these nanoparticles on the neuronal cell line.

The MTS assay showed that LiNbO_3_ nanoparticles did not affect the viability of PC12 cells, but instead increased it with treatments at 5.8 nM for 24 h and at 100 and 500 nM at all times tested. These data are contrast with those obtained in a previous study [21], where LiNbO_3_ nanoparticles negatively affected the PC12 cell viability. It is known that the degree of cell differentiation influences the behaviour of treated cells [23,53]. The higher cytocompatibility of LiNbO_3_ nanoparticles obtained in this study could be explained by the different medium used for cell differentiation, which could have modified the response of PC12 to treatment by enhancing their neuronal features.

Concerning the analyses of neuronal outgrowth, the results showed that treatment with LiNbO_3_ nanoparticles did not affect the differentiation grade or the number and length of neurites in PC12 cells: indeed, the neurite length was significantly increased by the treatment at 5.8 nM for 24 h. These data are consistent with those of a previous study analysing the effect of iron oxide nanoparticles on PC12 cell line [54]. The authors found an increased neurite outgrowth with an enhanced neurite number and length, confirming that metal ions have neuromodulatory effects on neurons, which could be exploited for the treatment of many neuronal diseases [54,55]. In addition, the number of branch points produced by PC12 cells was significantly enhanced by treatment with LiNbO_3_ at 100 and 500 nM, 48 h after treatment, and at 5.8 nM starting from 48 h after treatment. These data agree with those of a previous study analysing the effects of PC12 cells seeded on gold nanoparticles and stimulated by alternating electricity [56], and with those of another study analysing the effects of PC12 seeded with BNNT stimulated by ultrasound [45]. Overall, these results confirm that the electric stimulation has a neuromodulatory effect on neurons [57,58] and increase the importance of developing new scaffolds coated with piezoelectric nanoparticles, for use in the treatment of neuronal diseases.

In conclusion, the results obtained in this study on barium titanate and lithium niobate suggest the biocompatibility of both piezoelectric nanoparticles on a PC12 neuronal cell line and are consistent with other studies showing that piezoelectric materials are able to stimulate neuronal outgrowth and neurite length [39,42,59]. This makes piezoelectric nanoparticles an excellent tool for neural repair and regeneration. Although more experiments are required, barium titanate and lithium niobate nanoparticles appear to be good candidates for improving the efficiency of new implantable hearing devices without damaging the neurons. Recovery from SNHL is difficult and the development of a self-powered CI is a relevant innovation that could profoundly change the way these patients are treated. The incorporation of piezoelectric nanoparticles into PVDF fibres could improve the efficiency of this polymer, making it possible to produce the new-age CI for cochlear stimulation.

## 4. Materials and Methods

### 4.1. Compounds

Barium titanate (BaTiO_3_) (pm 233.192 g/mol, CAS Number 12047-27-7, Product Number: 467634) nanopowder and lithium niobate (LiNbO_3_) (pm 147.846 g/mol, CAS Number 12031-63-9, Product Number: 254290) powder, were purchased from Sigma-Aldrich (Milan, Italy). Before use, the nanoparticles were suspended in sterile-filtered 1% gelatine/double distilled water solution, and sonicated for 24 h in Elma Transsonic Ultrasonic Bath T 460 (35 kHz) (Elma Schmidbauer GmbH, Singen, Germany), to obtain well-dispersed nanoparticle suspensions. Cisplatin (cis-diamminedichloridoplatinum (II)) (Cpt) 3.3 mM was purchased from Accord Healthcare (Milan, Italy).

### 4.2. Cell Cultures

The PC12 cells, obtained from Interlab Cell Line Collection (ICLC ATL98004, Genoa, Italy), were grown in 5% CO_2_ at 37 °C in complete RPMI medium (Biowest, Nuaillé–France) supplemented with 10% Horse Serum (HS) (CARLO ERBA Reagents, Cornaredo, Milan, Italy), 5% Fetal Bovine Serum (FBS) (Euroclone, Pero, Milan, Italy), 2 mM L-Glutamine (Biowest), 1% penicillin/streptomycin (Sigma-Aldrich, Milan, Italy). To obtain neuron-like cell line, before treatment, the cells were plated and then differentiated for 6 days by a differentiation medium, chosen according to Hu and collaborators [60], Opti-MEM™ I Reduced Serum Medium (ThermoFischer, Milan, Italy), supplemented with 0.5% FBS, 1% penicillin/streptomycin and 50 ng/mL NGF (Sigma-Aldrich, Milan, Italy).

### 4.3. Cell Viability Assay

Cell viability was measured by CellTiter 96^®^ AQueous MTS Reagent Powder (Promega, Milan, Italy), according to the manufacturer’s suggestions. The PC12 cells were seeded 5000 cells/well in 6-well plates with 2 mL of complete medium and then differentiated for 6 days with differentiation medium. Once differentiated, the cells were treated with BaTiO_3_ and LiNbO_3_ at concentrations 5.8 nM, 100 nM and 500 nM. Cell viability was analysed 24, 48 and 72 h after treatment; untreated cells were used as negative control (NT) while cells treated with 13 and 25 µM cisplatin were used for positive control, according to previous studies [53,61,62]. At the end of each time interval, 100 μL of 3-(4,5-dimethylthiazol-2-yl)-5-(3-carboxymethoxyphenyl)-2-(4-sulfophenyl)-2H-tetrazolium (MTS) and phenazine metasulphate (PMS) mixture (20:1) were added to each well, containing 500 μL of fresh differentiation medium. The plates were incubated for 3 h at 37 °C. Absorbance was measured at 492 nm with a plate reader (SIRIO, SEAC srl, Florence, Italy).

### 4.4. Cell Morphology

Morphological alterations in cytoskeleton and nuclei were evaluated using phalloidin–tetramethylrhodamine (TRITC)/4′, 6-diamidino-2-phenylindole (DAPI) staining. The PC12 cells were seeded in 6-well plates on glass slides, cultured and treated as previously described. Untreated cells were used as negative control while cells treated with 13 µM cisplatin were used as positive control. At the end of each treatment, the cells grown on slides were fixed with Shandon™ Glyo-Fixx™ (ThermoScientific, Milan, Italy) for 30 min, washed 3 times in phosphate-buffered saline (PBS) (Sigma Aldrich, Milan, Italy) 1X and stained for 2 h in the dark with phalloidin–TRITC 1 μg/mL (Sigma Aldrich, Milan, Italy). After two washes in PBS 1X, the cells were stained with 1 μM DAPI (Sigma Aldrich, Milan, Italy) for 5 min in the dark. Finally, after two washes with PBS 1X, the slides were mounted with 20% glycerol (CARLO ERBA), and observed under the optical fluorescence microscope Nikon Eclipse TE2000-U (Nikon, Milan, Italy). The images were acquired using 40X magnification with the NIS element software (Nikon).

### 4.5. Cytochrome C Expression

The PC12 cells were seeded on glass slides, cultured and treated as previously described. At the end of each time interval, the cells were fixed in Shandon™ Glyo-Fixx™ for 30 min, washed 3 times in PBS 1X, incubated 30 min at 37 °C with rabbit serum in PBS 1X and incubated overnight at 4 °C with anti-cytochrome c primary antibody (sc13560, Santa Cruz Biotechnology, Dallas, Texas, USA). The next day, the samples were washed twice with PBS 1X and incubated 1 h in the dark with a TRITC-conjugated secondary antibody (Sigma Aldrich, Milan, Italy). After washing with PBS 1X, the cell nuclei were stained in blue by DAPI and the slides were mounted as previously described. The images were acquired at 40X magnification.

### 4.6. ROS Production

Levels of reactive oxygen species (ROS) were detected in PC12 cells using the H_2_DCFDA assay (Sigma Aldrich, Milan, Italy). The 2′-7′-dichlorodihydrofluorescein diacetate (H_2_DCFDA) is deacetylated in living cells to 2′, 7′- dichlorodihydrofluorescein (H_2_DCF), whose oxidation by ROS produces the fluorescent 2′, 7′-dichlorofluorescein (DCF). The fluorescence generated was proportional to the amount of H_2_DCF oxidized to DCF [63]. The PC12 cells were seeded in 96-well plates at 1000 cells/well, differentiated for 6 days as previously described and treated with nanoparticles. Untreated cells were used as negative control and cells treated with 25 µM cisplatin for 24 h were used as positive control. At the end of each time interval, the cells were treated with H_2_DCFDA 10 µM for 30 min at 37°C in the dark and after washing with PBS 1X, the fluorescent emission was measured at 485 nm excitation and 540 nm emission by Mithras LB 940 Multimode Microplate Reader (Berthold Technologies GmbH, Bad Wildbad, Germany).

### 4.7. Neurite Outgrowth

The PC12 cells were seeded in 6-well plates at 5000 cells/well, differentiated and treated with nanoparticles as previously described. Untreated cells were used as negative control and cells treated with 13 µM cisplatin as positive control. At the end of each time interval, fifteen different fields of view were acquired for treatment at 10X magnification with the inverted microscope Eclipse TE2000-U (Nikon). The images were analysed using the ImageJ software (https://imagej.nih.gov/ij/(accessed on 10 December 2021)). The parameters measured were the following:The differentiation grade, the ratio between the number of cells with neurites and the total cell number;The number of neurites per cell;The average length of neurites, the ratio between the total length of neurites and the number of cells with neurites;The number of branch points, the ratio between the number of branch points and the number of neurites per cell.

### 4.8. Statistical Analyses

Each test was performed at least three times in triplicate. One-way analysis of variance (ANOVA) or Kruskal–Wallis were used to assess the differences between multiple groups, followed by Student’s *t*-test or Mann–Whitney test. The *p*-value (*p*) <0.05 was considered statistically significant. All analyses were performed by the software GraphPad Prism 8.0.1 (https://www.graphpad.com/(accessed on 10 December 2021)).

## Figures and Tables

**Figure 1 ijms-23-01761-f001:**
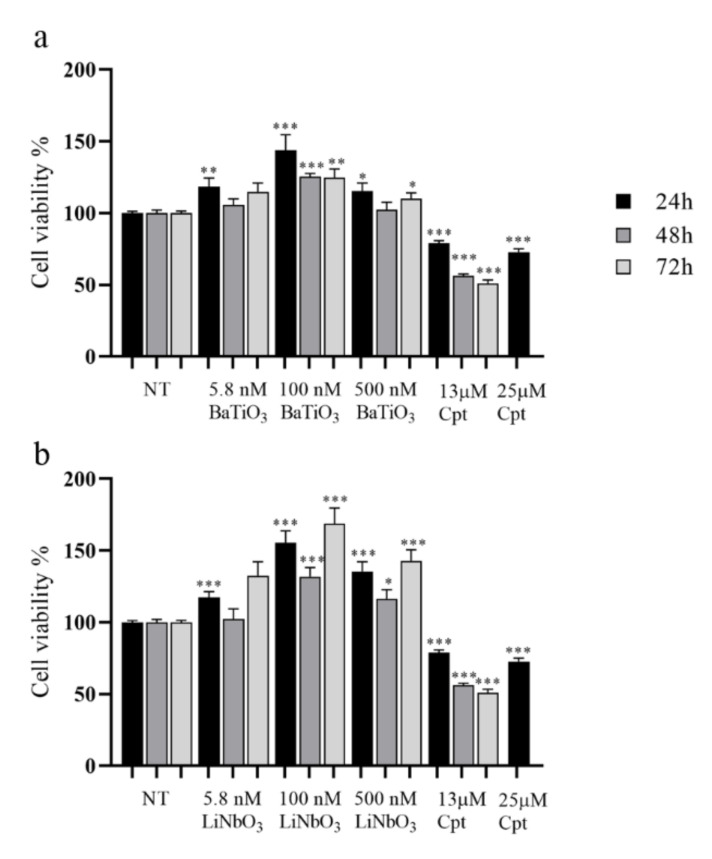
Results of MTS assay of differentiated PC12 cells treated with three different concentrations of barium titanate (BaTiO_3_) (**a**) or lithium niobate (LiNbO_3_) (**b**) for 24 h, 48 h and 72 h. Cell viability was expressed as mean value percent ± SEM *vs.* control cells (NT). Cisplatin (Cpt) 13 µM and 25 µM were used as positive control. Asterisks indicate significant differences in comparison to negative control (NT). * = *p* < 0.05, ** = *p* < 0.01, *** = *p* < 0.001.

**Figure 2 ijms-23-01761-f002:**
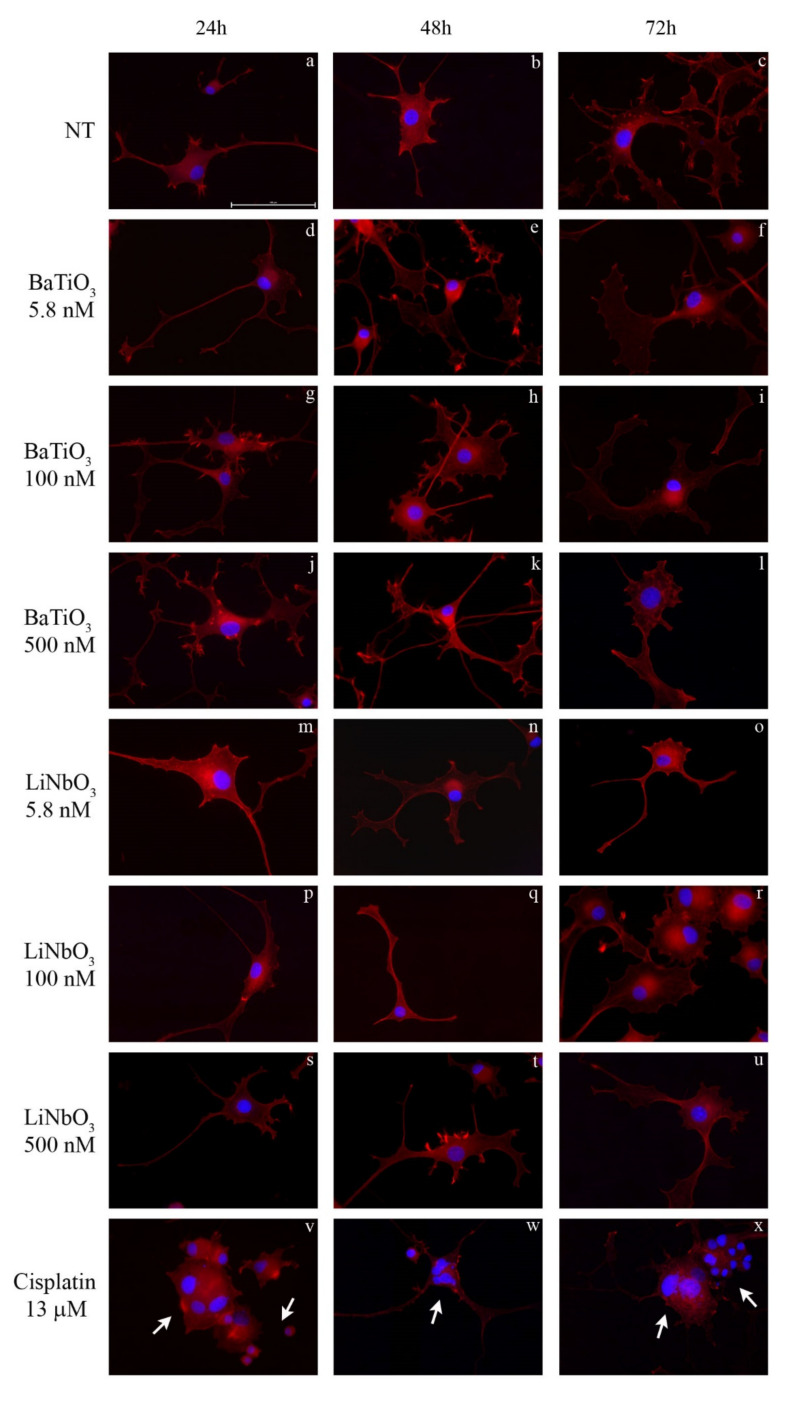
Morphological analysis of differentiated PC12 cells treated with BaTiO_3_ (**d**–**l**) or LiNbO_3_ (**m**–**u**) for 24 h, 48 h and 72 h, respectively. Untreated cells were used as control (**a**–**c**) and 13 µM cisplatin was used as positive control (**v**–**x**). The nuclei were stained in blue by DAPI, the cytoskeleton was stained in red by phalloidin-TRITC. Scale bar 100μm. Arrows indicate apoptotic cells.

**Figure 3 ijms-23-01761-f003:**
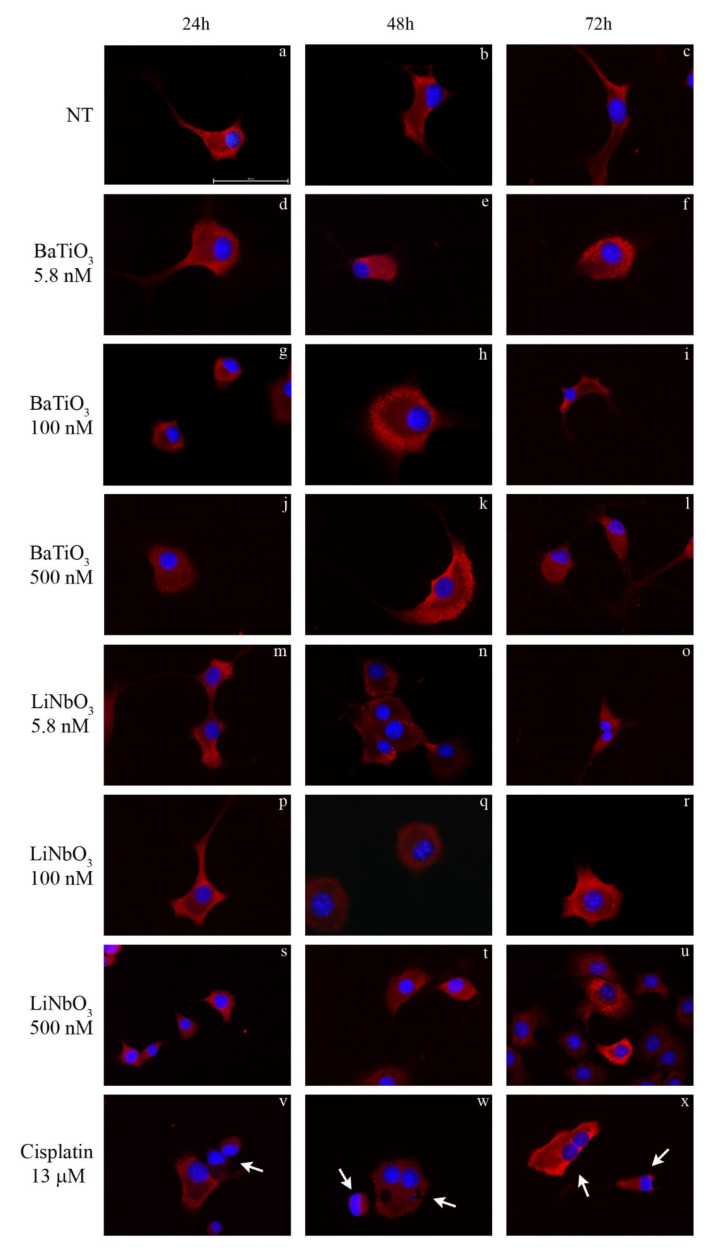
Immunocytochemical analysis of differentiated PC12 cells treated with BaTiO_3_ (**d**–**l**) or LiNbO_3_ (**m**–**u**) for 24 h, 48 h and 72 h. Untreated cells were used as control (**a**–**c**) and 13 µM cisplatin was used as positive control (**v**–**x**). Cytochrome c was stained in red by TRITC-conjugated secondary antibody; the nuclei were stained in blue by DAPI. Healthy cells show homogeneous cytochrome c staining while apoptotic cells show diffuse staining distribution. Scale bars 100 μm. Arrows indicate apoptotic cells.

**Figure 4 ijms-23-01761-f004:**
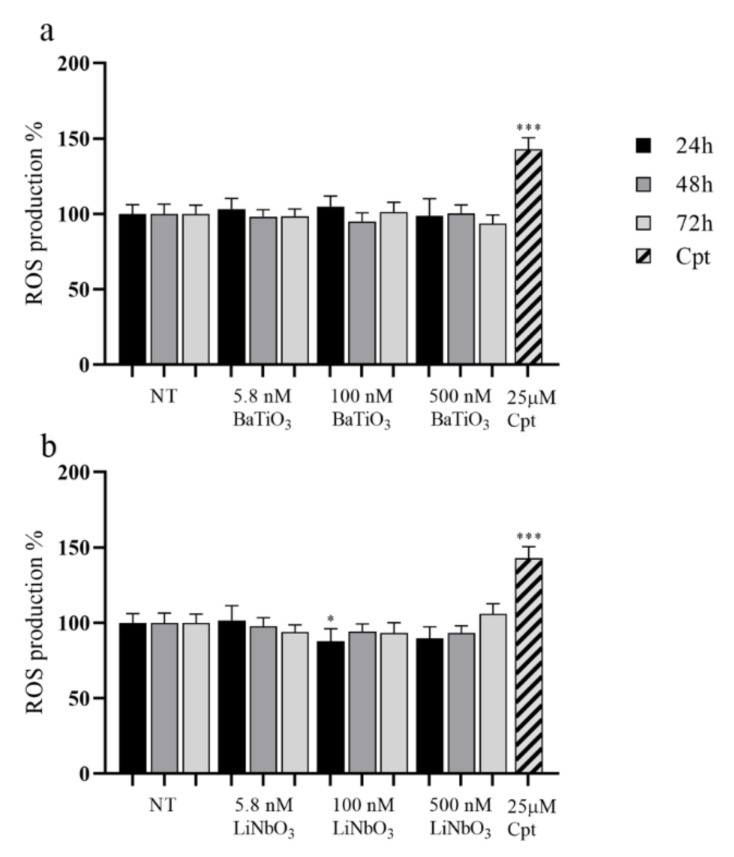
Results of H_2_DCFDA assay of differentiated PC12 treated with BaTiO_3_ (**a**) or LiNbO_3_ (**b**) for 24 h, 48 h and 72 h. Samples treated with 25 µM cisplatin (Cpt) for 24 h were used as positive control. The results are expressed as mean value percent ± SEM *vs.* control cells (NT). Asterisks indicate significant differences in comparison to NT. * = *p* < 0.05, *** = *p* < 0.001.

**Figure 5 ijms-23-01761-f005:**
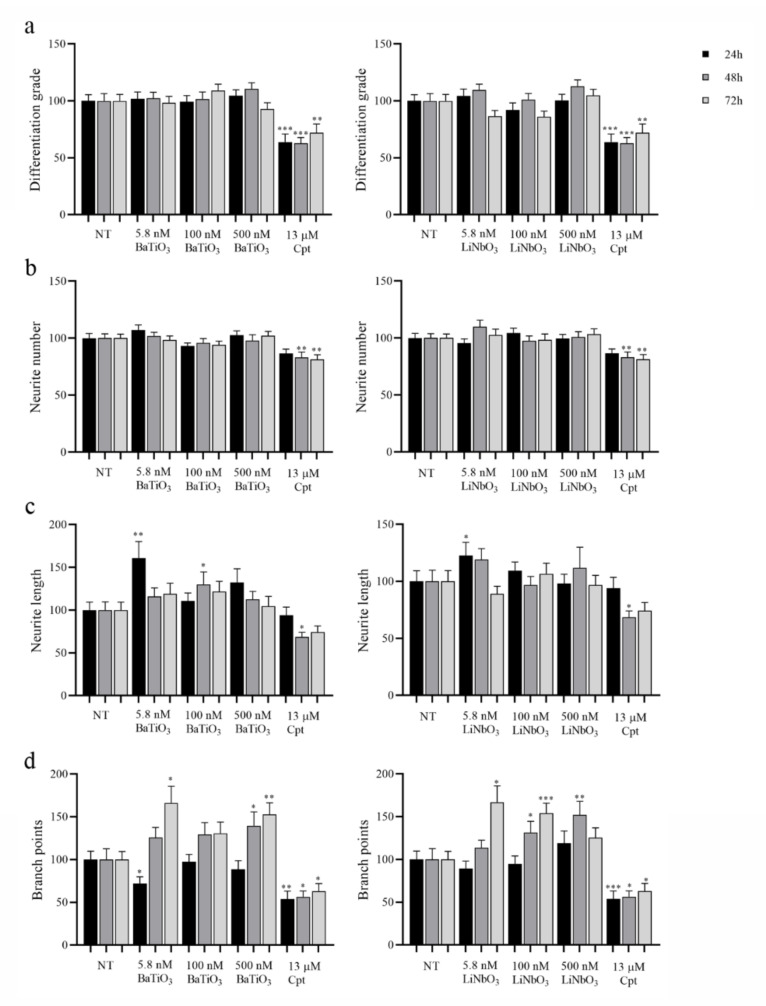
Evaluation of neurite network produced by differentiated PC12 cells treated with BaTiO_3_ or LiNbO_3_ for 24 h, 48 h, and 72 h. Cells treated with 13 µM cisplatin were used as positive control (Cpt). After incubation, 30 fields per treatment were acquired at magnification 10X and analysed measuring the differentiation grade (**a**), the number of neurites per cell (neurite number, (**b**)), the average length of neurites expressed in µm (neurite length, (**c**) and the number of branch points (**d**)). Data were expressed as mean value percent ± SEM vs. control cells (NT). Asterisks indicate significant differences in comparison to NT. * = *p* < 0.05, ** = *p* < 0.01, *** = *p* < 0.001.

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
