# Peer review of "Neuron Compatibility and Antioxidant Activity of Barium Titanate and Lithium Niobate Nanoparticles"

_ijms, 2022, doi:10.3390/ijms23031761_

Round 1
Reviewer 1 Report
In this paper, the authors reported the biocompatibility and the antioxidant activity of barium titanate and lithium niobate NPs on a neuronal cell line to explore the possible treatment of inner ear diseases, avoiding damages to neurons. The results showed that barium titanate and lithium niobate NPs do not affect viability, morphological features, cytochrome c distribution and ROS production by cells, and stimulate neurite branching.
This manuscript is recommended for publication upon addressing the following points:
- Abstract is general, missing important results and conclusions. Authors should provide the full description of the terms once and then use the abbreviation in the abstract. The used methods are missing in this section
- Introduction: missing the novelty and hence aim of the study.
- Since the study is about the treatment of neuronal cells via nanoparticles. There is no information about the Nanoparticles. Authors should provide the full description and detailed about the nanoparticles.
- The size, shape and structure of the nanoparticles is very important for any kind of biological activities. Considering the importance of these properties, the authors should provide the characterization data, especially the transmission electron microscopy (TEM) and/or scanning electron microscopy (SEM/EDX) of barium titanate and lithium niobate NPs even though the nanomaterials are commercial.
- The authors should provide the clear scale bars for figures 2 and 3.
- Since there are so many abbreviations are used, the authors should provide the list of abbreviation at the end of the MS.

Author Response
Reviewer n.1 In this paper, the authors reported the biocompatibility and the antioxidant activity of barium titanate and lithium niobate NPs on a neuronal cell line to explore the possible treatment of inner ear diseases, avoiding damages to neurons. The results showed that barium titanate and lithium niobate NPs do not affect viability, morphological features, cytochrome c distribution and ROS production by cells, and stimulate neurite branching. This manuscript is recommended for publication upon addressing the following points: - Abstract is general, missing important results and conclusions. Authors should provide the full description of the terms once and then use the abbreviation in the abstract. The used methods are missing in this section ANSWER: We thank the Reviewer for his/her valuable remark. Indeed, we added a sentence (highlighted in yellow) describing the methods and we provide the full description of the terms once and then use the abbreviation in the abstract section. - Introduction: missing the novelty and hence aim of the study. ANSWER: We thank the Reviewer for his/her comment. We add few sentences (highlighted in yellow) useful to emphasize the novelty and hence aim of the study in the introduction section. - Since the study is about the treatment of neuronal cells via nanoparticles. There is no information about the Nanoparticles. Authors should provide the full description and detailed about the nanoparticles. The size, shape and structure of the nanoparticles is very important for any kind of biological activities. Considering the importance of these properties, the authors should provide the characterization data, especially the transmission electron microscopy (TEM) and/or scanning electron microscopy (SEM/EDX) of barium titanate and lithium niobate NPs even though the nanomaterials are commercial. ANSWER: We thank the reviewer for his/her relevant advice. Actually, we have reported the procedure and nanoparticle characterization in previous studies. Anyway, we agree that this information is important to understand the study. According to the Reviewer’s suggestion, we described in more detail the nanoparticles adding the required information in the appropriate sections. In the revised version of our manuscript, we have now clearly reported that we refer to formulations characterized in previous works: “To investigate the cytocompatibility of BaTiO3 and LiNbO3 nanoparticles on PC12 cell line, the nanoparticles were treated according to an established procedure to generate a uniform dispersion [21]” and “In previous studies, we have established methodologies for obtaining suitable dispersions of these ceramic nanoparticles, which make them suitable for cellular uptake”. We have also added these supporting references of ours: “Ciofani G. et al.Nanoscale Research Letters, 2010, 5(7), pp. 1093–1101” and “Ciofani G. et al. Colloids and Surfaces B: Biointerfaces, 2010, 76(2), pp. 535–543”. Finally, SEM micrographs of the two particle types have been included as a supplementary material (Figure S1) and a statement was added in the Results section: “BaTiO3 and LiNbO3 had submicrometric size, as detected via scanning electron microscopy (SEM) analysis (Figure S1)”. - The authors should provide the clear scale bars for figures 2 and3. ANSWER: We thank the Reviewer for his/her comment and improved the quality of the scale bars in figures 2 and 3. - Since there are so many abbreviations are used, the authors should provide the list of abbreviation at the end of the MS. ANSWER: We thank the Reviewer for his/her suggestion, we provided the list of abbreviation here enclosed; we ask the Editor if we are allowed to insert this paragraph at the end of the MS, since it is not a section provided by the editorial guidelines. Abbreviations Barium titanate (BaTiO3); Lithium niobate (LiNbO3); Reactive oxygen species (ROS); Sensory Neural Hearing Loss (SNHL); Cochlear Implant (CI); polyvinylidene fluoride (PVDF); Nerve Growth Factor (NGF); Tetramethylrhodamine (TRITC); 4',6-diamidino-2-phenylindole (DAPI); 2',7'-dichlorodihydrofluorescein diacetate (H2DCFDA); the boron nitride nanotubes (BNNT); Horse Serum (HS); Fetal Bovine Serum (FBS); negative control (NT); (3-(4,5-dimethylthiazol-2-yl)-5-(3-carboxymethoxyphenyl)-2-(4-sulfophenyl)-2H-tetrazolium (MTS); phenazine metasulphate (PMS); phosphate buffered saline (PBS); 2’, 7’- di-chlorodihydrofluorescein (H2DCF); 2’, 7’-dichlorofluorescein (DCF).
Reviewer 2 Report
The submitted manuscript deals with the problem of the biocompatibility of the piezoelectric nanoparticles usable in the treatment of inner ear diseases. The presented results not only confirm good biocompatibility of the studied materials with neuronal cell line PC12 but also a positive effect on the growth of these cells which can boost additional research aimed at real application of these nanoparticles in the medicinal practice. The text is written in a good manner, all experimental results are sufficiently documented and discussed. Only several minor problems can be found in the text. The first is a technical problem with using subscript in the chemical formulae of the tested materials. The second problem is connected with insufficient characterization of tested nanomaterials – some prove of nanodimenzional character of these must be included in the text. Electron microscopic images are the best prove of this but also results of measurements obtained by some indirect method (DLS, XRD) will be usable.
Due to above mentioned comments to the text of the submitted manuscript I can recommend it for the publication in the International Journal of Molecular Sciences after minor revision.
Author Response
Reviewer n.2
The submitted manuscript deals with the problem of the biocompatibility of the piezoelectric nanoparticles usable in the treatment of inner ear diseases. The presented results not only confirm good biocompatibility of the studied materials with neuronal cell line PC12 but also a positive effect on the growth of these cells which can boost additional research aimed at real application of these nanoparticles in the medicinal practice. The text is written in a good manner, all experimental results are sufficiently documented and discussed.
- Only several minor problems can be found in the text.
ANSWER: We thank the Reviewer for his/her comment and we revised the editing highlighting in yellow the corrections.
- The first is a technical problem with using subscript in the chemical formulae of the tested materials.
ANSWER: We thank the Reviewer for his/her remark and corrected the chemical formulae
- The second problem is connected with insufficient characterization of tested nanomaterials – some prove of nano-dimensional character of these must be included in the text. Electron microscopic images are the best prove of this but also results of measurements obtained by some indirect method (DLS, XRD) will be usable.
ANSWER: We thank the reviewer for noticing this problem and suggesting possible solutions. Actually, we have reported the procedure and nanoparticle characterization in previous studies. Anyway, we agree that this information is important to understand the study and according to the Reviewer’s suggestion, we described in more detail the nanoparticles adding the required information in the appropriate sections.SEM micrographs of the two particle types have been included as a supplementary material (Figure S1) and a statement was added in the Results section: “BaTiO3 and LiNbO3 had sub-micrometric size, as detected via scanning electron microscopy (SEM) analysis (Figure S1)”. In the revised version of our manuscript, we have also clearly reported that we refer to formulations well characterized in previous works: “To investigate the cytocompatibility of BaTiO3 and LiNbO3 nanoparticles on PC12 cell line, the nanoparticles were treated according to an established procedure to generate a uniform dispersion [21We have also added these supporting references of ours: “Ciofani G. et al. Nanoscale Research Letters, 2010, 5(7), pp. 1093–1101” and “Ciofani G. et al. Colloids and Surfaces B: Biointerfaces, 2010, 76(2), pp. 535–543”.
Due to above mentioned comments to the text of the submitted manuscript
I can recommend it for the publication in the International Journal of Molecular Sciences after minor revision
Round 2
Reviewer 1 Report
Reviewer is satisfied with the responses of the authors.